

# Comparative analysis of flavonoids, polyphenols and volatiles in roots, stems and leaves of five mangroves

Zhihua Wu[1,2], Xiuhua Shang[1], Guo Liu[1] and Yaojian Xie[1]

[1] Research Institute of Fast-growing Trees, Chinese Academy of Forestry, Zhanjiang, Guangdong, China
[2] School of Forestry, Nanjing Forestry University, Nanjing, Jiangsu, China

## ABSTRACT

Mangrove plants contain a variety of secondary metabolites, including flavonoids, polyphenols, and volatiles, which are important for their survival and adaptation to the coastal environment, as well as for producing bioactive compounds. To reveal differences in these compounds among five mangrove species' leaf, root, and stem, the total contents of flavonoids and polyphenols, types and contents of volatiles were determined, analyzed and compared. The results showed that *Avicennia marina* leaves contained the highest levels of flavonoids and phenolics. In mangrove parts, flavonoids are usually higher than phenolic compounds. A total of 532 compounds were detected by a gas chromatography-mass spectrometry (GC-MS) method in the leaf, root, and stem parts of five mangrove species. These were grouped into 18 classes, including alcohols, aldehydes, alkaloids, alkanes, *etc.* The number of volatile compounds in *A. ilicifolius* (176) and *B. gymnorrhiza* (172) was lower than in the other three species. The number of volatile compounds and their relative contents differed among all three parts of five mangrove species, where the mangrove species factor had a greater impact than the part factor. A total of 71 common compounds occurring in more than two species or parts were analyzed by a PLS-DA model. One-way ANOVA revealed 18 differential compounds among mangrove species and nine differential compounds among parts. Principal component analysis and hierarchical clustering analysis showed that both unique and common compounds significantly differed in composition and concentration between species and parts. In general, *A. ilicifolius* and *B. gymnorrhiza* differed significantly from the other species in terms of compound content, while the leaves differed significantly from the other parts. VIP screening and pathway enrichment analysis were performed on 17 common compounds closely related to mangrove species or parts. These compounds were mainly involved in terpenoid pathways such as C10 isoprenoids and C15 isoprenoids and fatty alcohols. The correlation analysis showed that the content of flavonoids/phenolics, the number of compounds, and the content of some common compounds in mangroves were correlated with their salt and waterlogging tolerance levels. These findings will help in the development of genetic varieties and medicinal utilization of mangrove plants.

Corresponding author
Yaojian Xie, cercxieyj@163.com

## INTRODUCTION

The growth and development of plants are often affected by incompatible environments such as drought, salt, cold, frost, and elevated temperatures that result in low yields and, in worse cases, the death of the plants (*Chen et al., 2022*). Abiotic stresses such as flooding, heat, drought, cold, *etc.*, and biotic stresses such as pathogenic attacks lead to the formation of some secondary metabolites, which play important roles in plant survival and create ecological connections between other species (*Jan et al., 2021*). Many secondary metabolites produced by medicinal plants have anti-microbial properties, high antioxidant levels, cytotoxic properties, as well as other properties that are medically quite significant (*Punetha et al., 2022*).

Mangrove is a unique forest ecosystem distributed in tropical and subtropical coastal areas (*Dahibhate, Saddhe & Kumar, 2019*). To adapt to harsh natural environments such as high salinity, high temperature, and low oxygen, mangrove plants have evolved into highly developed morphological and physiological adaptability (*Dahibhate, Saddhe & Kumar, 2019*). The secondary metabolites of mangroves are unique and novel with diverse bioactive functions (*Das, Samantaray & Patra, 2016*), which enable them to endure biotic and abiotic stresses and adapt to harsh environmental conditions (*Dahibhate, Saddhe & Kumar, 2019*; *Sivaramakrishnan et al., 2019*). In addition, mangrove plants are rich in flavonoids, steroids, terpenes, alkaloids, and other chemical compounds (*Dahibhate, Saddhe & Kumar, 2019*; *Karim et al., 2021*). These bioactive and natural compounds may be used as precursors for pharmaceuticals and industrial raw materials (*Dahibhate, Saddhe & Kumar, 2019*). As one of the important medicinal plants, mangrove plants are widely used as traditional (ethnic) in the world (*Karim et al., 2021*). Mangrove plants have been reported to contain active chemical ingredients useful for medicinal purposes in the past few decades. Structural types and the biological activities of natural products found in true mangroves and semi-mangroves worldwide have been summarized (*Wu et al., 2008*; *Li et al., 2009*). The current reports on the activity detection of compounds from mangrove plants indicate that mangrove plants are a valuable source of pharmacologically active substances with broad prospects (*Okla et al., 2019*).

However, plants have developed several mechanisms to counteract the effects of abiotic stress at the morphological, anatomical, biochemical, and molecular levels, including changes in secondary metabolite production due to exposure to environmental stress (*Punetha et al., 2022*). Metabolomics is considered a fundamental branch of systems biology (*Rosato et al., 2018*). It provides a powerful tool for understanding abiotic stresses in plants and developing resistance strategies at the metabolite level (*Carrera et al., 2021*). Gas chromatography-mass spectrometry (GC-MS) is one of the most effective, reproducible, and widely used analytical platforms because of its robustness, repeatability, and selectivity of the technology and a large number of mature commercial and metabolite databases. GC-MS has become an important method of choice for metabolomic analysis and answering various biological questions in metabolomics (*Feizi et al., 2021*). Recently, studies on mangrove ecosystems have focused on microorganisms associated with mangroves rather than on flora (*Wu, Xu & Guo, 2022*). Although GC-MS has also been

applied to the detection of different bioactive compounds in mangrove plant extracts (*Bidve, Kadam & Malpathak, 2018*; *Joel & Bhimba, 2010*; *Lalitha et al., 2021*; *Dahibhate & Kumar, 2022*; *Swaraiah et al., 2020*; *Huang et al., 2019*), fewer studies have applied metabolomics to analyze and compare metabolic differences among root, stem, and leaf parts of different mangrove species. Under stress conditions, different metabolites are allocated or synthesized in different parts of plants; however, little is known about how genotypic differences affect these processes (*Kang et al., 2019*). In this study, the GC-MS techniques of volatile compounds were established for five mangrove species. The flavonoids, polyphenols, and volatiles of their leaves, stems, and roots were analyzed and compared, aiming to understand the structures, biosynthesis, and resources of these metabolites in five mangrove species with different adaptations.

## MATERIALS & METHODS

### Plant material

The seedlings of five mangroves, *Acanthus ilicifolius* L. (*genus* Acanthus), *Bruguiera gymnorrhiza* L. (*genus* Bruguiera), *Aegiceras corniculatum* L. (*genus* Myrsinaceae), *Kandelia candel* L. (*genus* Kandelia), and *Avicennia marina* F. (*genus* Avicennia) were collected from Tongming river, which is located in Zhanjiang Mangrove National Nature Reserve (ZMNNR), Guangdong (E110.1667°, N 20.9765°). These species are the most common native mangrove species in the ZMNNR area, and have been identified and recorded in the national mangrove resource survey in 2001 (*Gao, Han & Liu, 2009*). Field experiments were approved by Guangdong Provincial Science and Technology Department and Guangdong Provincial Forestry Bureau (project number: 20208020214001). The seedlings of the five mangrove species were 1.5 years old and 25–50 cm high (Fig. 1), and were identified by ZMNNR Professor Yuechao Chen. Three plants of each mangrove species were used for biological duplication, and their leaves, roots, and stems were taken. The combinations of species and organ parts from the leaves, roots, and stems were coded and named species_part. Leaves, roots, and stems of *A. ilicifolius* (AI), *B. gymnorrhiza* (BG), *A. corniculatum* (AC), *K. candel* (KC), and *A. marina* (AM) were simplified as AI_leaf, AI_root, AI_stem, BG_leaf, BG_root, BG_stem, AC_leaf, AC_root, AC_stem; KC_leaf, KC_root, KC_stem, AM_leaf, AM_root, AM_stem, respectively.

### Determination of flavonoids and polyphenols

The total flavonoid content was determined by the protocol described previously (*Kim et al., 2003*). The absorbance of flavonoids was determined at 506 nm using a visible spectrophotometer (PE Lambda-6, Waltham, MA, USA). Based on the linear equations (y = 1.2051x−0.0066 ($R^2$ = 0.9988, SEE (the sum of squares due to error) = 0.0119)) from catechin (purity >97.50%, Macklin Inc, Shanghai) standard solutions of different concentrations, flavonoid concentrations were calculated. Spectrophotometric analysis was performed to determine the content of total phenolics/polyphenols using Folin-Ciocalteu's phenol reagent (*Kim et al., 2003*). The absorbance of polyphenols was also determined using a PE Lambda-6 spectrophotometer at 765 nm, and the polyphenol content was calibrated using a linear equation (y = 7,1266x + 0.0095 ($R^2$ = 0.9993, SEE = 9,070)) from
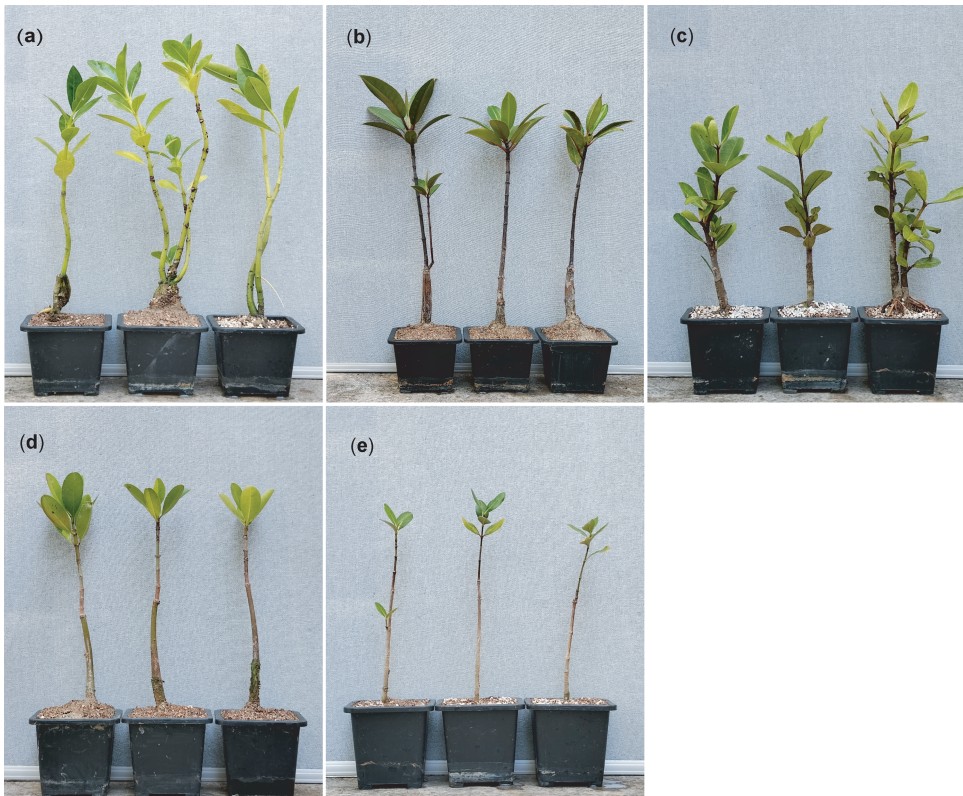

**Figure 1** **Morphological characteristics of the seedlings of five mangrove species used in this study.** (A) *A. ilicifolius*; (B) *B. gymnorrhiza*; (C) *A. corniculatum*; (D) *K. candel*; (E) *A. marina*. The pot with a square mouth has the following specifications: the upper pot side length is 4.2 cm, the bottom side length is 3.8 cm, and the pot height is 4.0 cm. The cultivation substrate is mainly composed of rice husk, coconut shell shreds, vermiculite, and mixed with sea mud. During the cultivation period, the seedlings are watered every 2–3 days with a salinity of about 0.3% seawater.

gallic acid (purity >99.00%, Macklin Inc, Shanghai) reference solution. As a percentage of the sample's dry mass, the final content of flavonoids or phenols was obtained.

## Sample preparation

In April 2020, healthy leaves, stems and roots from the whole plant of each mangrove species were picked, washed and dried in the shade and crushed. A total of 5 g of leaves, roots, and stems, respectively, were weighed accurately and put into a triangular flask. Then, a total of 100 mL of 70% ethanol was added and extracted at 25 °C for 48 h (shaken for 5 min every 12 h), and then centrifuged for 10 min at 4,000 R/min. The supernatant was filtered with quantitative filter paper and concentrated to dryness under reduced pressure, then 1 mL of ethyl was added to dissolve and prepared to test.

## GC-MS analysis

A total of 1.0 μL of the sample solution was used in a Shimadzu GC-2010 gas chromatograph (Shimadzu Scientific Instrument, Inc., Columbia, MD, USA).

The GC-MS system was equipped with an HP-5ms (30 m × 250 µm × 0.25 µm) chromatographic column, made of (5%-phenyl)-methylpolysiloxane.

In the heating procedure, the initial temperature was set at 40 °C for 2 min, increased at 5 °C/min to 230 °C and held at 230 °C for 2 min, then increased to 250 °C at 20 °C/min and held at 250 °C for another 2 min. Samples were injected in splitless mode; Injection time was 1.00 min. The carrier gas was helium with a flow rate of one mL/min. The temperature of the GC injector was 250 °C.

The GC MS-QP2010 SE mass spectrometer was operated in EI mode at 70 eV of electron energy; ion source temperature was equal to 230 °C, and the interface temperature was equal to 250 °C. The solvent delay time was 1.00 min; The scan interval was 0.30 s with a 2,000 amu/s scan speed, and the scan Mas range was 50~550 m/z. The maximum length of retention time was 43.00 min. All measurements were repeated three times.

## Calibration curve of standard solutions for GC-MS

Three methyl esterified standards (Sigma-Aldrich, St. Louis, MO, USA) were used for the qualitative analysis by external standard method. Separately, a certain amount of methanolic linoleic acid, methanolic octadecanoic acid and methanolic linolenic acid standards were weighed and prepared as 0.1, 0.2, 0.3, 0.4 and 0.5 mg/ml of n-hexane, concentration gradient solution each. Six standards, including 3-octanol, limonene, 3-cyclohexen-1-ol, hexadecane, cedrol and heptadecane, were supplied by Sigma-Aldrich and were selected for validation of study findings. Six standards were prepared as 0.04, 0.06, 0.08% (v/v) n-hexane concentration gradient solution (where cedrol concentration is 0.04, 0.06, 0.08 mg/mL) with reference to the above method. Each solution was used for GC-MS analysis, and the measurements were repeated three times. The mean values were calculated. The regression equations between the peak areas and the concentrations of the standards solutions were obtained as follows:

$$y_{\text{linolenicacid}} = -3.65547 \times 10^6 + 7.66475 \times 10^9 x, (R^2 = 0.99336)$$
$$y_{\text{octadecanoicacid}} = 9.84569 \times 10^5 + 9.65142 \times 10^8 x, (R^2 = 0.98193)$$
$$y_{\text{linoleicacid}} = 2.60555 \times 10^8 + 6.71208 \times 10^{10} x, (R^2 = 0.99835)$$
$$y_{3-\text{octanol}} = -1.74715 \times 10^7 + 1.07282 \times 10^9 x, (R^2 = 0.9131)$$
$$y_{\text{limonene}} = -8.13324 \times 10^6 + 1.83422 \times 10^9 x, (R^2 = 0.9995)$$
$$y_{3-\text{cyclohexen}-1-\text{ol}} = -3.79158 \times 10^7 + 2.37430 \times 10^9 x, (R^2 = 0.8630)$$
$$y_{\text{hexadecane}} = 7.03236 \times 10^7 + 4.06271 \times 10^9 x, (R^2 = 0.9364)$$
$$y_{\text{cedrol}} = 4.94046 \times 10^6 + 4.65335 \times 10^8 x, (R^2 = 0.9379)$$
$$y_{\text{heptadecane}} = 5.02313 \times 10^7 + 5.02804 \times 10^9 x, (R^2 = 0.9905)$$

## Data analysis

The mass spectral fragmentation patterns of the compounds detected by GC-MS were compared with those in the NIST 2014, Wiley (version 9) libraries. Those with a mass spectral similarity of over 90% were selected for identification. The relative content (%)
of each compound was calculated by comparing the peak area, expressed as % from total peak areas, in GC-MS analysis. PLS-DA (partial least squares discriminant analysis) is a statistical technique for feature extraction and supervised discriminant analysis (*Barker & Rayens, 2003*), which produces a nice separation to distinguish the observed values between groups (*Barker & Rayens, 2003*). A relationship model between the metabolomic data block and the respective labelled groups was established by PLS-DA, which could detect the variables that influence the differences between groups and classify and predict samples. VIP (variable important in projection) is a PLS-DA weight value (*Banerjee et al., 2013*), which can be used to measure the influence intensity and explanatory ability of the accumulation difference of each metabolite for the classification and discrimination of each group of samples (*Tang et al., 2021*). In the PLS-DA models, the VIP parameter was used to identify metabolites that make the most contribution to diagnostic group discrimination, and threefold cross-validation of the models was conducted to evaluate their predictive ability (*Kaddurah-Daouk et al., 2013*). The prcomp function of R language (R version 4.2.0, (2022-04-22), *R Core Team, 2022*) and PCA of the ggplot2 package was used for research and visualization. Scatterplot and pheatmap packages were used to prepare the corresponding to scatter plots and heatmaps of compounds. The upset analysis diagram was performed using TBtools software (*Chen et al., 2020*). MetaboAnalyst (http://www.metaboanalyst.ca) is a network application platform for metabolomic data analysis and interpretation and other omics association analysis (*Kaddurah-Daouk et al., 2013*). The data were processed successively as follows, first removing the features with more than 50% missing values and then the missing values replaced by LoDs (1/5 of the minimum positive value of each variable). To eliminate batch-to-batch differences, the data were normalized by Pareto scaling. The statistical module of MetaboAnalyst 5.0 was used to carry out a one-way statistical analysis of mangrove compounds from species and parts, IDs of the metabolic compound were searched in an HMDB database (Human Metabolome Database, https://hmdb.ca/) and a PubChem database (https://pubchem.ncbi.nlm.nih.gov/), and obtained the corresponding KEGG (Kyoto Encyclopedia of Genes and Genomes, https://www.genome.jp/kegg/) IDs, and pathway enrichment analysis was performed in 1,072 subchemical class metabolome or lipome dataset.

# RESULTS

## Flavonoid and polyphenol contents of five mangroves

Figure 2 shows that AM_leaf contains the highest level of flavonoids with a value of 12.19%, followed by BG_root and KC_root. Additionally, AM_leaf contains the highest concentration of phenolics, reaching 7.87%. However, the average distribution of phenolics was about 2% in other parts of mangrove species. In general, the flavonoid content of different parts of mangroves was higher than the phenolic content. As a result of clustering based on flavonoid and phenolic content, AM_leaf and BG_root clustered into one group, which markedly distinguished different parts of other mangroves. However, AM_root and AM_stem exhibited relatively low flavonoid and phenolic contents, which were also noticeably different from those of other mangrove species. The same organ parts, such as the leaf, were generally clustered into one subclass.

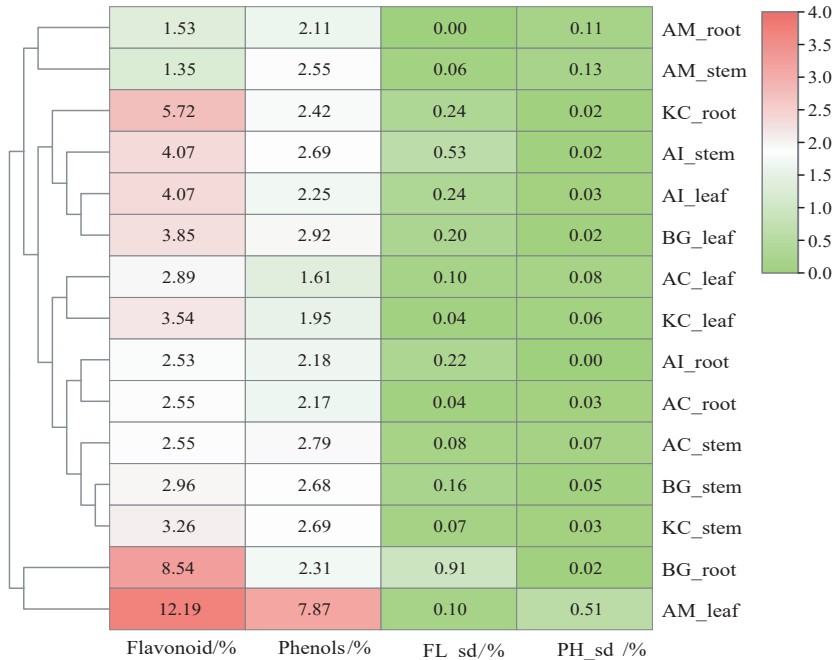

| Flavonoid/% | Phenols/% | FL_sd/% | PH_sd /% | |
|---|---|---|---|---|
| 1.53 | 2.11 | 0.00 | 0.11 | AM_root |
| 1.35 | 2.55 | 0.06 | 0.13 | AM_stem |
| 5.72 | 2.42 | 0.24 | 0.02 | KC_root |
| 4.07 | 2.69 | 0.53 | 0.02 | AI_stem |
| 4.07 | 2.25 | 0.24 | 0.03 | AI_leaf |
| 3.85 | 2.92 | 0.20 | 0.02 | BG_leaf |
| 2.89 | 1.61 | 0.10 | 0.08 | AC_leaf |
| 3.54 | 1.95 | 0.04 | 0.06 | KC_leaf |
| 2.53 | 2.18 | 0.22 | 0.00 | AI_root |
| 2.55 | 2.17 | 0.04 | 0.03 | AC_root |
| 2.55 | 2.79 | 0.08 | 0.07 | AC_stem |
| 2.96 | 2.68 | 0.16 | 0.05 | BG_stem |
| 3.26 | 2.69 | 0.07 | 0.03 | KC_stem |
| 8.54 | 2.31 | 0.91 | 0.02 | BG_root |
| 12.19 | 7.87 | 0.10 | 0.51 | AM_leaf |

**Figure 2   Content heatmap of flavonoids and phenols of five mangrove species and parts.** In the figure, $FL_{sd}$ represents the flavonoid standard deviation, and $PH_{sd}$ represents the polyphenol standard deviation. The color scale in the legend is on a log2 scale from 0–4. The data in the heatmap are the percentages in the dry sample. The clustering dendrogram of samples was based on their Euclidean distance using the complete clustering method.

## Volatile comparison of five mangrove species

The total ion current (TIC) diagrams of leaves, stems, and roots of five mangrove species detected by GC-MS, were shown in Fig. 3 and Fig. S1. There are 532 different compounds in the mangrove parts, involving 18 classes, including alcohols, aldehydes, alkaloids, alkanes, *etc.* (Fig. 4, Tables S1, S2). From Fig. 4, it can be seen that some mangrove parts contained large amounts of alkane-like compounds, followed by acids, then alcohols, ketones and esters, especially alkanes were widely distributed and abundant in the whole plant of *A. marina, A. corniculatum,* and *B. gymnorrhiza* species. In addition, the roots and stems of *K. candel* and *B. gymnorrhiza* were rich in monocyclic aromatic compounds.

Figures 5–8 showed that a minimum of 41 compounds (in BG_stem) and a maximum of 86 compounds in AC_leaf and AM_root could be determined in each part of five mangrove species. Furthermore, the more compound parts were the *K. candel* root with 85 compounds detected, and *A. corniculatum* root with 84 compounds detected.

A total of 247 compounds were found in *A. corniculatum*, 244 in *K. candel*, and 240 in *A. marina* (Fig. 5A). The number of compounds in three mangrove species was significantly more than that in *A. ilicifolius* (176) and *B. gymnorrhiza* (172). The roots of mangrove plants contained the greatest number of compounds (399), followed by the stems (341), and finally the leaves (339). Upset analysis of these compounds was studied, and unique and common compounds among different sample groups were identified (Fig. 5B). There

Peer J

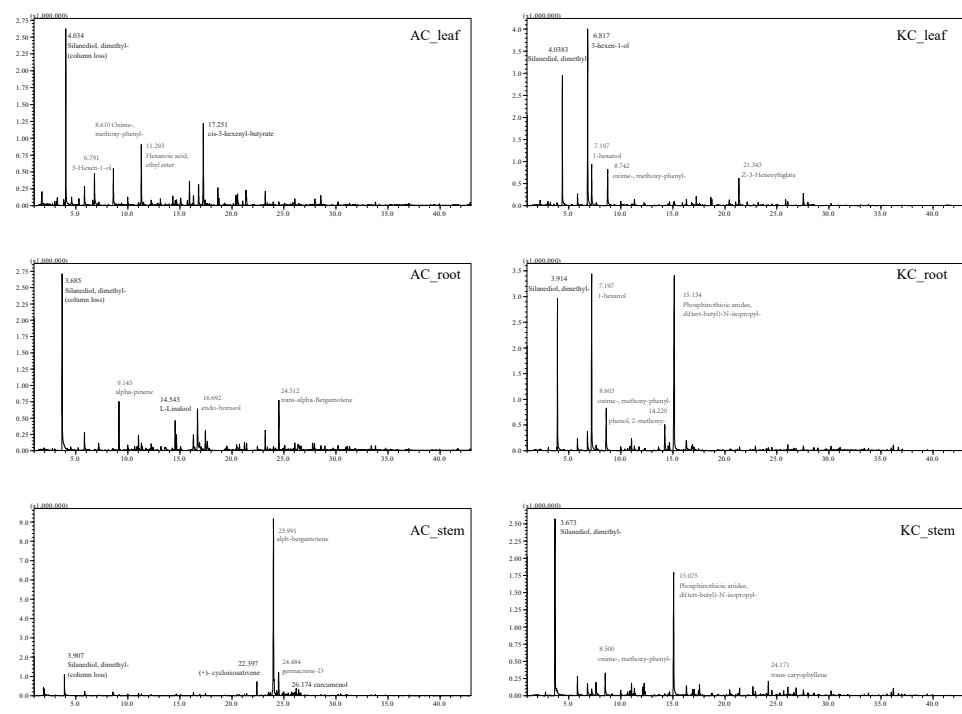

**Figure 3** **The total ion current (TIC) diagram of two mangrove leaves, stems, and roots by GC-MS.**
The number and compound marked next to the peak in the figure indicate the retention time of the compounds, respectively. At 3-4 min of retention time, a certain amount of column loss was present.

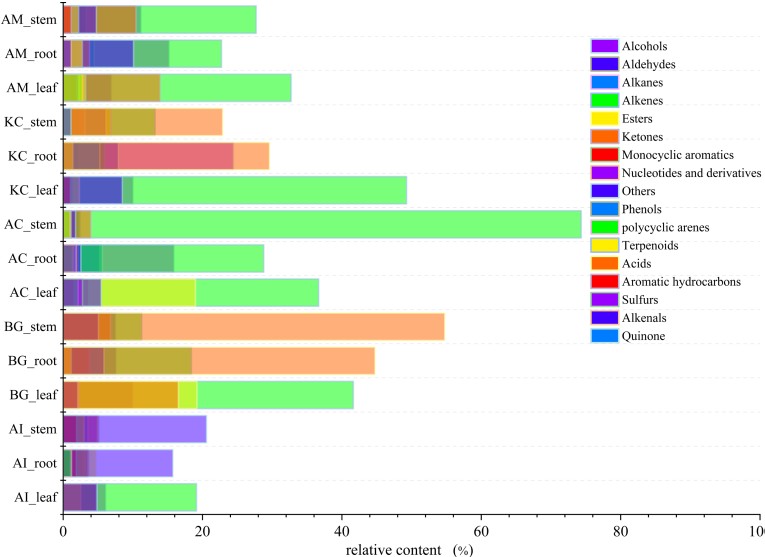

**Figure 4** **The distribution and content of 18 classes of compounds in different mangrove species and parts.**

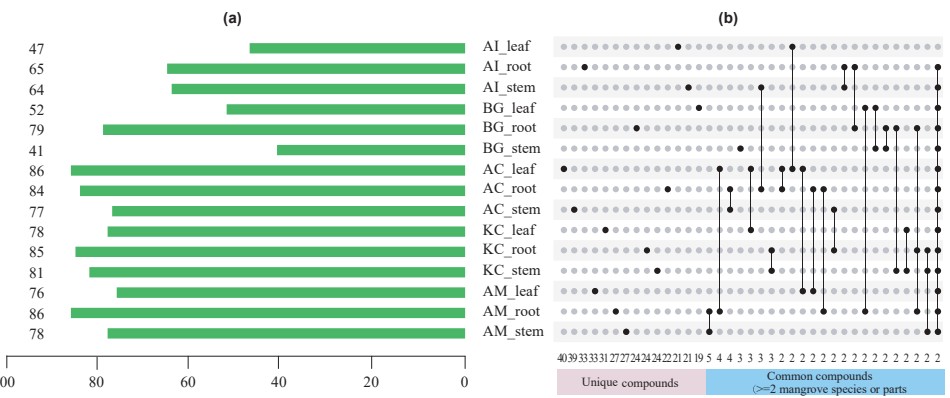

**Figure 5 The number of chemical compounds in leaves, stems, and roots of five mangrove species by GC-MS.** (A) The number of compounds tested in different species and parts. The number on the left side of the column is the number of compounds tested, (B) distribution of unique or common compounds on different mangrove species and parts. The number indicates the number of unique compounds in this column (only one marked black dot in the unique compound group) or the number of common compounds containing >2 in this column (multi-dot connected by a line in the common compound group). The details of the compounds in each combination of mangrove species and parts are referred to the Dryad database (https://doi.org/10.5061/dryad.ffbg79cz8).

were 40 unique compounds found in *A. corniculatum* leaves, 39 in *A. corniculatum* stems, 33 in *A. ilicifolius* roots and *A. marina* leaves, and just three in *B. gymnorrhiza* stems. There was a 30.16% high concentration of 3-hexen-1-ol, acetate (z) - in *B. gymnorrhiza* leaves (BG_leaf), followed by acetic acid (E) -non-3-enyl ester (9.17%) in *A. marina* roots (AM_root) and hexanoic acid, ethyl ester (8.1%) in *A. corniculatum* leaves (AC_leaf), followed by endo-borneol (AC_root) and 2-heptenoic acid, ethyl ester (AI_leaf) (Fig. 6).

The term "common compound" refers to a chemical compound found in multiple parts of 2 or more mangrove species (Fig. 5B). The most common compounds were found in the *A. marina* roots and stems, with five common compounds.

The total content of the compounds detected (Fig. 7A) and five mangrove species' unique compounds (Fig. 7B) were compared. The highest levels of detected compounds in *B. gymnorhiza* leaves (92.67%), followed by *A. corniculatum stems* (90.39%), and *B. gymnorhiza* stems (87.54%). *B. gymnorrhiza* had the highest percentage at 88.58% among mangrove species. Leaves had the highest content among the three parts.

Two factors were used to estimate the total compound content and unique compound content of mangrove species (factor A) and parts (factor B). $R^2$ of the two models was greater than 0.999 with *P* values less than 0.001 (Tables S3, S4). The regression equations of the two models were shown as follows:

$$\text{Total content of detected compounds} = 70.74 - 26.57A_{[1]} + 17.85A_{[2]} + 7.54A_{[3]} \\ + 5.36A_{[4]} + 0.8581B_{[1]} - 1.35B_{[2]} - 0.4955A_{[1]}B_{[1]} + 3.23A_{[2]}B_{[1]} - 2.88A_{[3]}B_{[1]} \\ - 1.12A_{[4]}B_{[1]} - 0.9991A_{[1]}B_{[2]} - 1.71A_{[2]}B_{[2]} - 8.76A_{[3]}B_{[2]} + 8.44A_{[4]}B_{[2]} \tag{1}$$
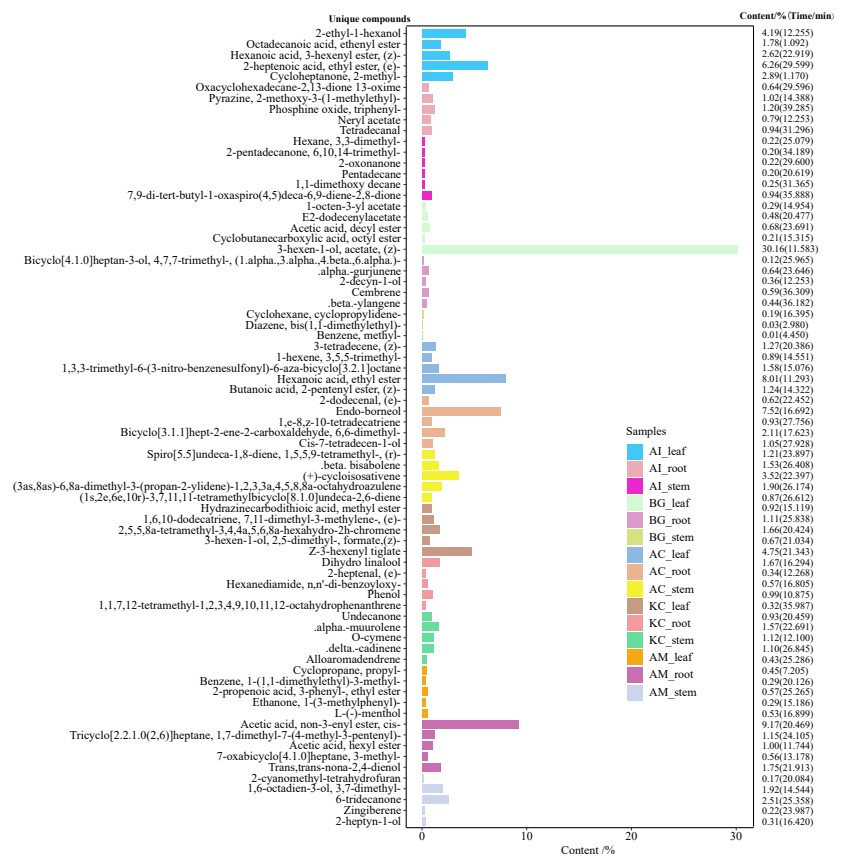

**Figure 6  Top 5 unique compounds ranked by content in five mangroves and parts.** The left vertical co-ordinate in the plot indicates the metabolite unique compound name. The horizontal coordinates indicate the relative content (%) and the right vertical coordinates indicate the relative content of the unique compounds and their retention time. The different colored bars in the figure indicate the different mangrove species and parts.

Total content of unique compounds $= 12.86 + 0.28A_{[1]} - 0.84A_{[2]} + 6.66A_{[3]}$
$-3.59A_{[4]} + 6.64B_{[1]} - 1.45B_{[2]} + 3.83A_{[1]}B_{[1]} + 14.01A_{[2]}B_{[1]} - 4.23A_{[3]}B_{[1]}$   (2)
$-2.39A_{[4]}B_{[1]} + 0.54A_{[1]}B_{[2]} - 7.39A_{[2]}B_{[2]} - 0.55A_{[3]}B_{[2]} - 2.09A_{[4]}B_{[2]}$

The regression (Eq. (1)) of the total content of the detected compounds indicates that the maximum coefficient for mangrove species is 26.57 from $A_{[1]}$ which suggests that mangrove species (factor A) had the greatest influence. $A_{[3]}$ and $B_{[3]}$ in Eq. (2) have maximum coefficients of 6.66 and 6.64, respectively, which indicates that mangrove species and parts have almost equal factor effects on the total content of unique compounds.

## Differences of compounds in three parts of mangrove plant

A total of 71 common compounds in five mangrove species were detected and used for further analysis. Figure S1 shows that *K. candel* leaves had the fewest common compounds, whereas their stems and roots had more common compounds. The content of most

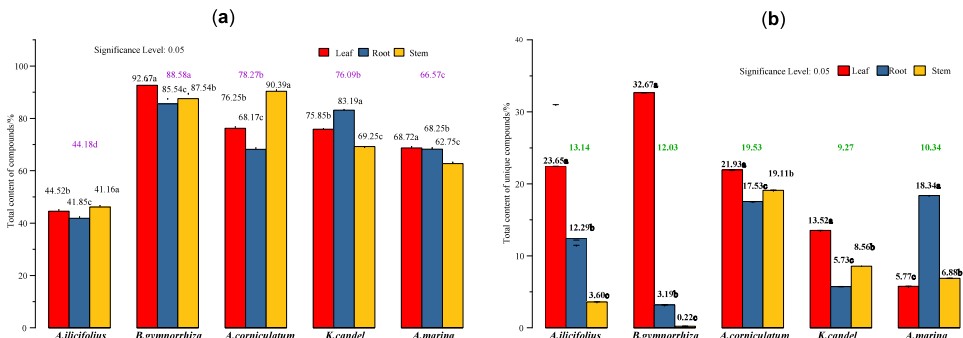

**Figure 7 Total content of various compounds in three parts of five mangrove species.** (A) Total content of detected compounds; (B) total content of unique compounds. In the figure, the numbers on the histogram are the total contents of compounds tested in different parts of five species, followed by different lowercase letters, indicating that the tukey comparison between different parts in the same species is significantly different at the level of 0.05. The color numbers and lowercase letters indicate the total means of species and their differences at the level of 0.05.

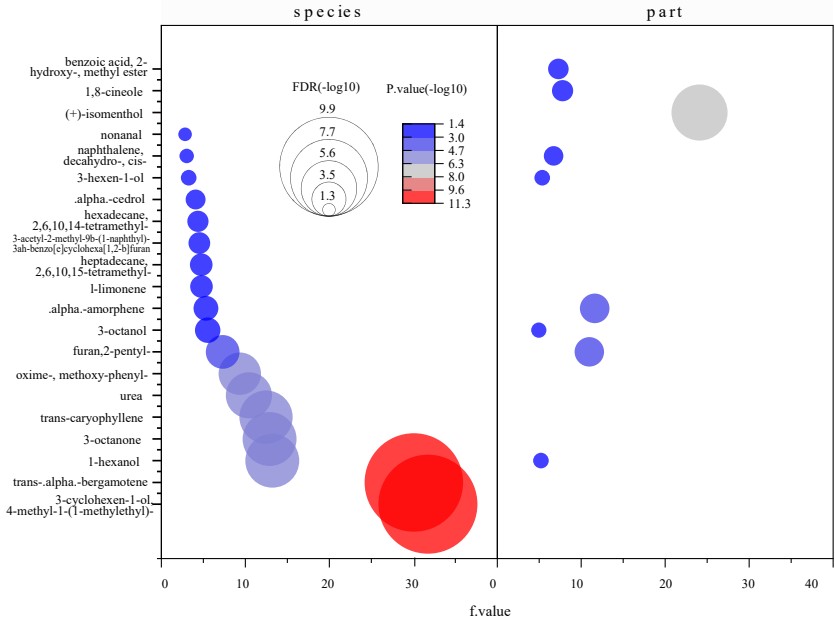

**Figure 8 Bubble diagrams of common compounds with significant differences based on species and parts.** *F* value, an F-statistic value computed by Fisher's LSD test, tested whether the effects of the different treatments were all equal, in the one-way ANOVA. FDR is an adjusted *p* value calculated by Benjamini–Hochberg method. The size of the circles in the plot indicates that the FDRs were processed by -log10 transformation, similarly, different colors and numbers in the color scale indicate the *p* value after -log10 preprocessing.

common compounds in the *A. ilicifolius* roots and stems, and *A. marina* leaves and stems were concentrated in a particular range, indicating no significant difference in their contents. It was noted that *B. gymnorrhiza* and *A. corniculatum* leaves contained a few

common compounds with high concentrations, which were primarily distributed at the 5-15 min retent time (Fig. S2), and the top five compounds were methoxy phenyl oxime, 1-hexanol, 3-octanone, 3-octanol, and 3-hexen-1-ol in order of their content.

A total of 71 common compounds were further analyzed using the metaboanalyst platform. Eighteen different compounds were found among mangrove species and nine among parts (Fig. 8). $\alpha$-Amorphene, furan 2-pentyl-, naphthalene 1-methyl-, 1-hexanol, and 3-octanol were the differential compounds in the two models. Statistically significant differences were found between species and parts with log10(P)>1.3 and FDR (false discovery rate) less than 0.05. These indicated significant differences in the contents of common compounds between mangrove plants.

## Statistical and pathway enrichment analysis of common compounds

VIP $\geq$ 1 was taken as the screening criterion, and the differences among groups were preliminarily screened out (*Tang et al., 2021*). The model prediction accuracy $Q^2$ (the cross-validated $R^2$) is calculated by cross-validation (CV) and has a high level of predictive ability when $Q^2$ is greater than 0.5 (*Triba et al., 2015*). When PCs (principal component) equals five in the PLS-DA model (Fig. 9), the accumulated $Q^2$ of species factor and part factor is 0.81 and 0.78, respectively, which indicates that the two models had high reliability (Fig. 9). The species factor model (Fig. 10A) has a contribution rate of 32.1% and 22.4% for component 1 and component 2, while the other part factor has a contribution rate of 42.4% and 7.4% for component 1 and component 2 (Fig. 10B). The samples of five mangrove species could be clearly separated on the two-dimensional plot of PLS-DA (Fig. 10A). There is a clear separation between the light green (*A. ilicifolius*) and the cool blue (*B. gymnorrhiza*) groups along the PC1 direction of Fig. 10A. PC1 is closely related to mangrove species because intra-species differences in common compounds are relatively minor among the three other mangrove species. Leaves (pink group) and stems (blue group) in Fig. 10B could be distinguished, despite the overlap of stems and roots. The findings of the clustering analysis of common compounds (Fig. S3) indicated that the leaves of *B. gymnorrhiza, A. corniculatum,* and *A. ilicifolius* formed a distinct class. Another big class consisted of three subclasses.

Further, VIP scores of 71 common compounds were discriminated against and sorted by two PLS-DA models (Fig. 11), with the top five being hexanol, 3-octanone, trans-caryophyllene, trans-alpha-octanone, and oximemethox in the species factor model, and L-limonene, (+) -isomenthol, 3-hexen-l-ol, 1-hxanol, and 3-octanone, naphthalene in the part factor model. Functional pathway enrichment analysis (Table 1 and Table S5) was performed with 17 differential common compounds identified by two PLS-DA models. As shown in Table 1, the results showed that 11 of the differential common compounds were enriched on six pathways, according to the enrichment ratio (−log10(P) value). These pathways were C10 isoprenoids, C15 isoprenoids, fatty alcohols, hydrocarbons, diterpenoids, and monoterpenoids.

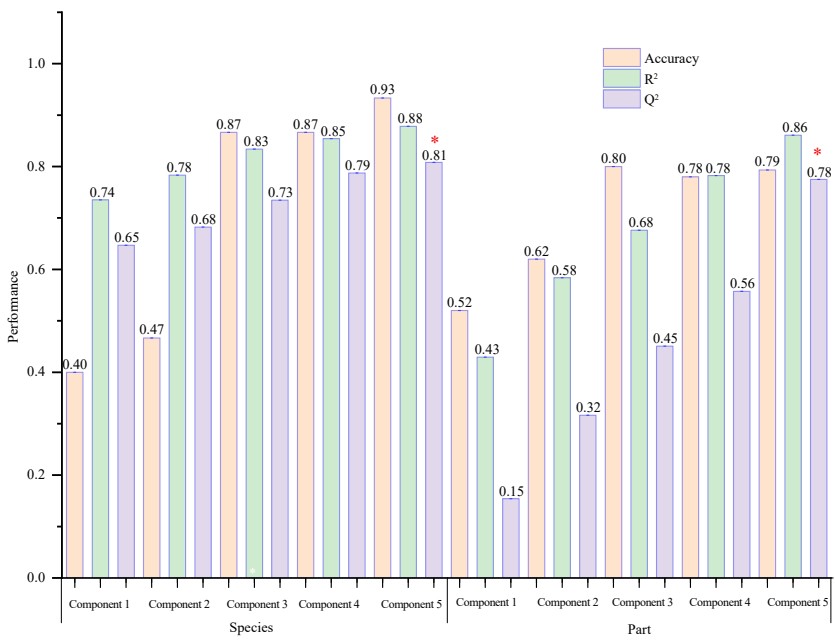

**Figure 9 A column chart of the PLS-DA cross-validation result.** The selected performance measure—$Q^2$ showed that the five-component model was best (indicated by a red star).

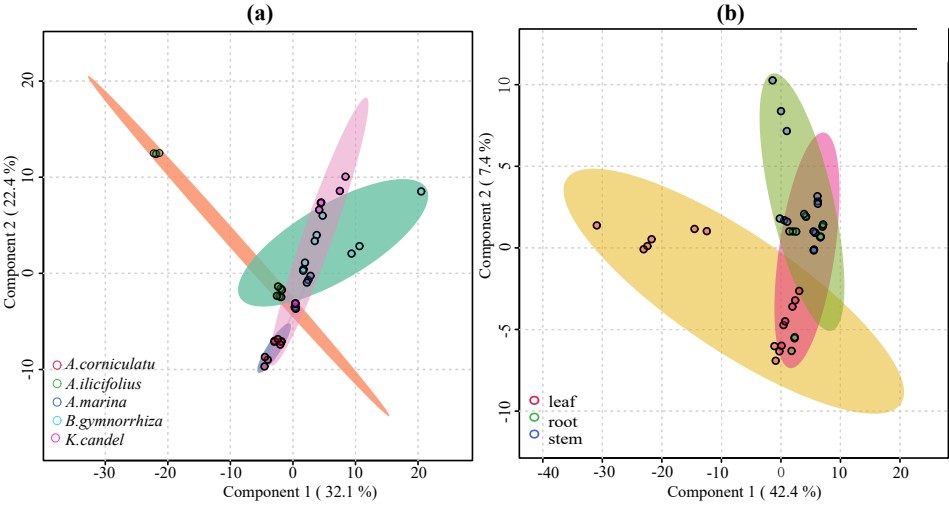

**Figure 10 2D score plot of the first two components from common compounds.** (A) Species factor; (B) part factor. The circles in the figure represent the various samples.

## DISCUSSION

### Role and differences of flavonoids and phenolics

Plant secondary metabolites, mainly phenolics, flavonoids, alkaloids, and terpenoids, are involved in environmental adaptation and stress tolerance (*Boncan et al., 2020*). Flavonoids

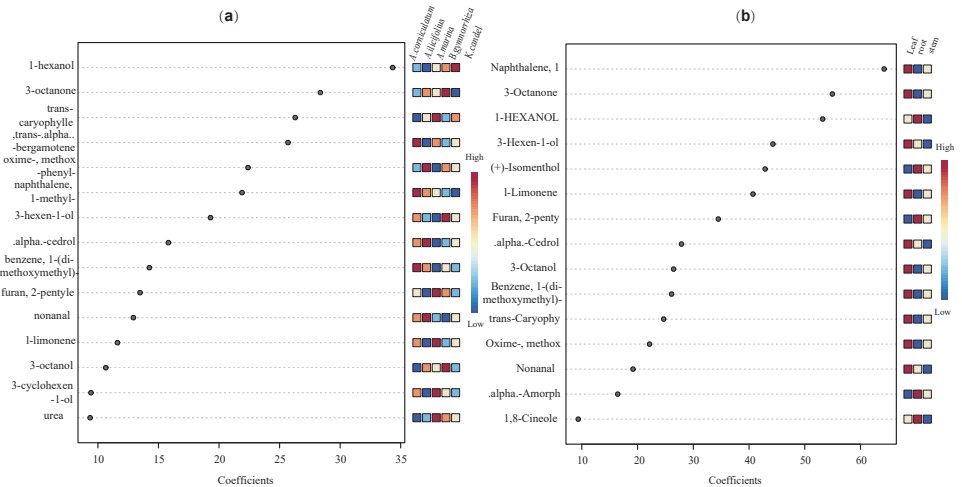

**Figure 11** **The VIP scores plot of the top 15 important compounds ranked by VIP coefficient.** (A) species factor, and (B) part factor.

**Table 1** **Metabolic pathway enrichment analysis of 17 common compounds in three parts of five mangrove specie.**

| Enrichment ratio (-log10(P)) | Metabolite set | Total | Hits | P value | Holm correction P value | FDR | Compounds |
|---|---|---|---|---|---|---|---|
| 5.9 | C10 isoprenoids | 242 | 3 | $1.20 \times 10^{-6}$ | 0.00126 | 0.00126 | eucalyptol, isomenthol, D-Limonene |
| 5.5 | C15 isoprenoids | 333 | 3 | $3.12 \times 10^{-6}$ | 0.00326 | 0.00163 | $\alpha$-Copaene, $\alpha$-amorphene, $\alpha$-cedrol |
| 3.2 | fatty alcohols | 452 | 2 | $6.87 \times 10^{-4}$ | 0.717 | 0.24 | 3-hexen-1-ol, 3-octanol |
| 2.4 | hydrocarbons | 52 | 1 | 0.00444 | 1 | 1 | heptadecane, 2, 6, 10, 15-tetramethyl |
| 1.6 | diterpenoids | 276 | 1 | 0.0234 | 1 | 1 | hexadecane, 2, 6, 10, 14-tetramethyl |
| 1.5 | monoterpenoids | 348 | 1 | 0.0294 | 1 | 1 | 3-cyclohexen-1-ol, 4-methyl-1-(1-methylethyl) |

and polyphenols are the most important secondary metabolites commonly distributed in the kingdom of plants (*Panche, Diwan & Chandra, 2016*) and play an essential role in plant growth, development, stress resistance, and other biological processes (*Panche, Diwan & Chandra, 2016*). Flavonoids participated in some plants' stress response process (*Samanta, Das & Das, 2011*; *Frank et al., 2021*), which involved flavonoid accumulation (*Samanta, Das & Das, 2011*). Mangrove plants are potential sources of medicines due to the presence of bioactive compounds. The antioxidant activity of mangrove plants was related to their phenolic and flavonoid contents (*Krishnamoorthy et al., 2011*). *Rhizophora apiculata* and *A. ilicifolius* root extracts were rich sources of phenolic compounds and flavonoids (*Asha, Mathew & Lakshmanan, 2012*). In this study, the flavonoid and phenolic contents of the five mangrove species were rich and different in the parts of the mangrove species. Flavonoid content was greater than phenolics, which differed significantly from the stem barks of *Bruguiera cylindrica* and *Ceriops decandra* in India (*Krishnamoorthy et al., 2011*). The results in Fig. 2 showed that among the five mangroves, AM_leaf contained the highest levels of flavonoids (12.19%) and the highest concentration of phenolics

(7.87%). The flavonoid and phenolic contents of five mangrove species and three parts were further correlated. The results (Fig. 12) showed that the flavonoid content and phenolic content of the same part were significantly correlated with large correlation coefficients (absolute values), such as flavonoids and phenolics in leaves with 0.99 correlation coefficients, and strong correlation between mangrove stems and leaves in flavonoid content or phenolic content. These suggested that the flavonoid and phenolic content of mangrove plants varied with different species and parts, which was influenced by age, origin, and ecological environment (*Amirav, Fialkov & Alon, 2013*). Thus, we assumed that these differences were not only from mangrove species and parts, but also from the mangrove environment. In natural conditions, mangrove species are zoned according to their adaptability, especially tolerance to flooding (*Chen et al., 2017*). Mangrove species in Leizhou Peninsula, Guangdong are *A. marina* → *A. corniculatum* → *K. candel* →*B. gymnorrhiza* →*A. ilicifolius* in order of distribution from the coast to the inner shore (*Chen et al., 2017*). Four mangrove species showed a decrease in flooding tolerance in the following order: *A. marina* >*A. corniculatum* >*Rhizophora stylosa* >*B. gymnorrhiza* (*He et al., 2007*). Compared to *K. candel, B. gymnorrhiza* was not as waterlogging tolerant (*Ye et al., 2003*). Therefore, we classified five mangrove species into five levels of salt and waterlogging tolerance from weak to strong. The correlation between the flavonoid or phenolic content of mangroves and their levels of salt and waterlogging tolerance was analyzed (Fig. 12). The results showed that salt and waterlogging tolerance levels were negatively correlated with the flavonoid/phenolic content of roots and stems, while the flavonoid/phenolic content of leaves was positively correlated. According to these results, flavonoids and phenolic content is highly correlated with the adaptation level of mangrove species.

## Differences in volatiles between mangrove species and their part

As a powerful and unique analytical method, GC-MS has become increasingly popular to analyze medicinal plants in recent years (*Al-Rubaye, Hameed & Kadhim, 2017*). We have developed and optimized GC-MS conditions to detect volatiles from the leaves, stems, and roots of five mangrove species based on previous studies (*Wu et al., 2012*). In this study, 532 compounds in three organ parts were identified. The number of compounds in each part identified by GC-MS ranged from 41 to 85, and was higher than that reported in some literature on mangroves (*Kumar Dinesh & Rajakumar, 2016*; *Joel & Bhimba, 2010*; *Lalitha et al., 2021*; *Dahibhate & Kumar, 2022*; *Swaraiah et al., 2020*). Furthermore, we observed that GC-MC rarely detected common fatty acids. There were two reasons for this; the first was that GC-MS was limited in its ability to analyze a limited number of volatile, thermally stable chemicals, and the second was that mangrove species rarely contain fatty acids. Only *A. ilicifolius* and *A. marina* have been reported to contain fatty acids (*Bibi et al., 2019*).

In this study, five mangroves contained 532 compounds, of which 341 compounds were found in stems, 339 in leaves, and 399 in roots (Fig. 5), and these differences were not significant among roots, stems, and leaves. There were also differences in the number of compounds among the five species. However, the number of compounds in all three parts and the total mangrove plant was also positively correlated with their salt and waterlogging

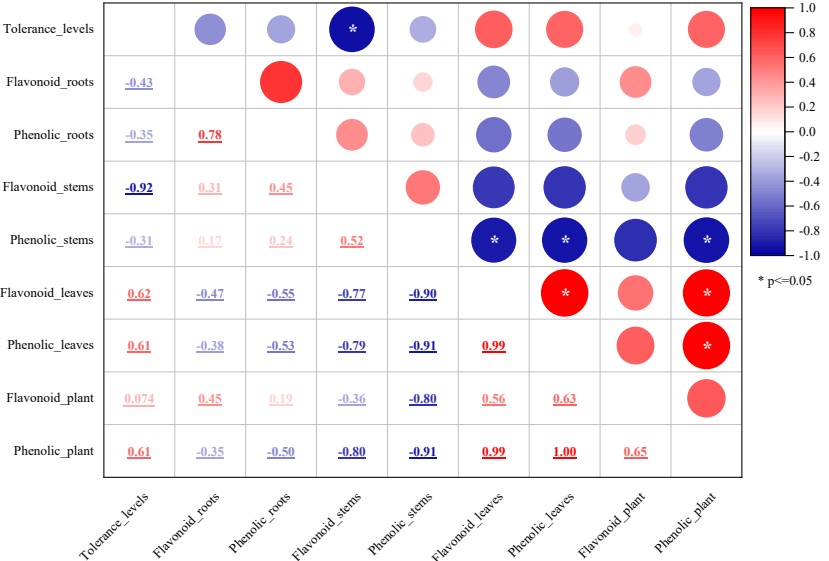

**Figure 12 The relationship between flavonoid or phenolic contents of mangroves and their levels of salt and waterlogging tolerance.** The mean values of flavonoid or phenolic contents in each part or whole plant of the five mangrove species. Variables of flavonoid or phenolic contents represented as compound name plus parts/whole plant, *e.g.*, Flavonpid_roots and Flavonpid_plant. Tolerance levels were evaluated from weak to strong on a scale of 1–5 level as follows: AI (level 1), BG (level 2), KC (level 3), AC (level 4), and AM (level 5).

tolerance levels (Fig. S4). Among these species, the total content of detected compounds differed significantly (Table S3), indicating that different tolerance mangrove species contain varying levels of compounds. The one-way analysis of 71 common compounds revealed that 18 metabolites that differed were found between mangrove species, while nine metabolites were found between parts. In summary, mangrove species had greater factorial effects on metabolite differences than part factors.

## Differential compounds involved in metabolic pathways and their role

Among these 532 compounds, 114 compounds with KEGG ID (Table S1) were mainly involved in the following three metabolic pathways (Fig. 13): monoterpenoid biosynthesis (Fig. 13 and Fig. S5), sesquiterpenoid and triterpenoid biosynthesis, and $\alpha$-Linolenic acid metabolism (Fig. 13). The metabolic pathway enrichment analysis of different common compounds was mainly involved in the metabolism of C10 isoprenoids, C15 isoprenoids, fatty alcohols, *etc.* (Table 1).

Terpenes, also known as terpenoids or isoprenoids, comprise the most chemically and structurally diverse family of natural products (*Christianson, 2017*). Their main skeleton comprises five-carbon isopentyl units, known as 2-methyl-1,3-butadiene or isoprene (*Yeshi et al., 2022*). Monoterpenoids are composed of two isoprene units (C-10 carbon atoms) and sesquiterpenoids of three isoprene units (C-15 carbon atoms) (*Yeshi et al., 2022*). The results of the above two analyses are consistent. Thus, our study suggested that the main

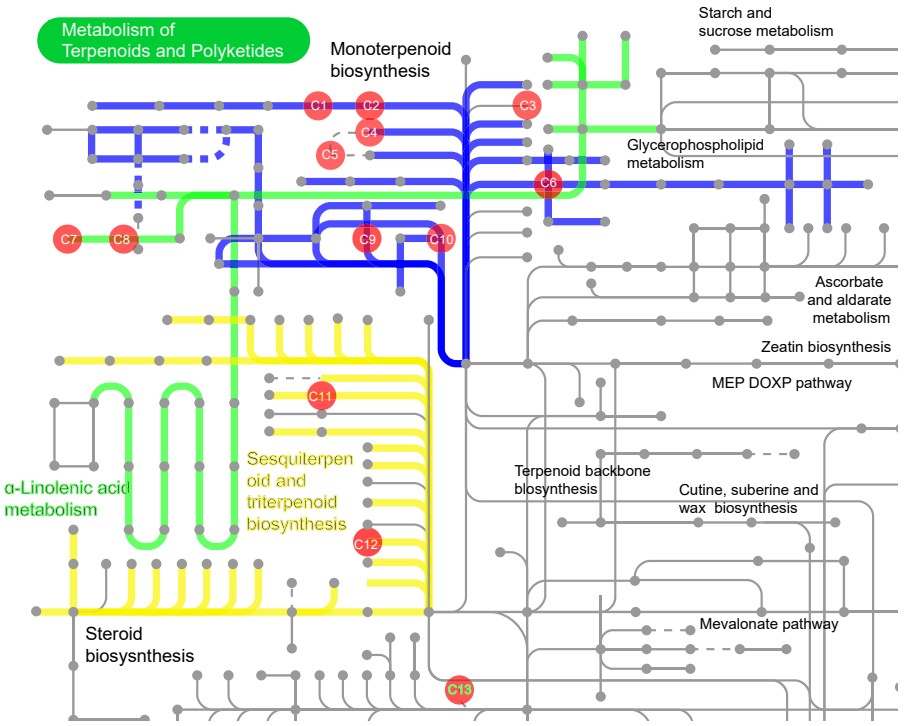

**Figure 13** **The volatile compounds with KEGG ID involved in secondary metabolic pathways in iPath3.0 (http://pathways.embl.de).** Red dots indicate compounds in the metabolic pathways. Seven compounds were found in the pathway of monoterpenoid biosynthesis, and two compounds each in the sesquiterpene and triterpene biosynthesis pathway, as well as in the $\alpha$-linolenic acid metabolism synthesis pathway. C1: 1,8-cineole (KEGG ID: C09844); C2: (-)-alpha-Terpineol (C11393); C3: (-)-3-Carene (C09839); C4: (-)-alpha-Pinene (C06308); C5: alpha-Pinene (C09880); C6: (-)-Limonene (C00521); C7: cis-3-Hexenyl acetate (C19757); C8: 3-Hexen-1-ol (C08492); C9: (-)-Linalool (C11388); C10: Myrcene (C06074); C11: (2E,6E)-Farnesol (C01126); C12: beta-Caryophyllene (C09629); C13: Cembrene (C11893).

differences in the GC-MS metabolites of the five mangroves and their parts were related to the isoprenoids or terpenoids.

Plant volatiles are involved in plant-environment interactions and, to some extent, in abiotic stress responses (*Dudareva, Pichersky & Gershenzon, 2004*; *Dudareva et al., 2006*). The volatile emission profile of plants may be one of the signature responses to stress conditions as plant volatiles mitigate the effects of oxidative stress (*Boncan et al., 2020*). Terpenoids play a defensive role in plants responding to biotic and abiotic stresses (*Yeshi et al., 2022*). Isoprene can aid plant performance under abiotic stresses (*Frank et al., 2021*). The isoprene can improve thermotolerance or reduce oxidative stress, preventing herbivores and parasitoids from attracting plants (*Sharkey, Wiberley & Donohue, 2008*). The triterpenoid content played an important role in mangrove plants for protection from salinity in both salt-secretors and non-secretors (*Basyuni et al., 2012*). *A. marina*'s metabolites and metabolic pathways were important factors contributing to its salinity and drought stress tolerance (*Ravi et al., 2020*). Concerning the tolerance levels of the

five species mentioned above, the common compounds that differed in the one-factor statistical analysis of mangrove species (Fig. 11A) were analyzed for correlation between the compound content and the tolerance levels of the species (Fig. S6). It was found that seven compounds in leaves had significant correlation with >0.6 absolute values of correlation coefficients, three of which reached significant levels. Among the three parts, the mean value of correlation coefficients (absolute values) was the largest in leaves. Again, these indicated that some compound content correlated with the tolerance level of mangrove species, especially the content of compounds in the leaves is more correlated.

Meanwhile, terpenoids have different biological activities (*Mitić et al., 2017*), including good antibacterial activity (*Patra & Mohanta, 2014*), cytotoxic and antiviral activity (*Gong et al., 2017*), and moderate cytotoxic and antimicrobial activities (*Cerri et al., 2022*). Terpenes and terpenoids are the main bioactive compounds of essential oils (*Masyita et al., 2022*), which have been comprehensively studied and reported to play critical roles in human health (*Perveen, 2018*). Mangrove metabolites have potential applications in the discovery of novel medicinal properties.

The above results suggested that different mangrove species have various secondary metabolite profiles associated with different tolerance levels of salt and waterlogging. Further related studies should be conducted to understand the precise mechanisms behind this relationship and identify specific secondary metabolites that contribute to mangrove resistance to salt and waterlogging. For bioactive secondary metabolites in mangroves, more attention should be paid to the abiotic stresses affecting their pharmacological properties.

## CONCLUSIONS

Mangrove plants grow in harsh coastal areas where interaction with the surrounding environment is inevitable. Their production of secondary metabolites, especially organic compounds and terpenes, is related to their ability to withstand biotic and abiotic stresses. In this paper, flavonoids, polyphenols, and volatiles in mangroves' roots, stems, and leaves were tested, analyzed, and compared. Flavonoid and phenolic contents varied among mangrove species and parts, which are highly correlated with the adaptation levels of mangrove species to salt and waterlogging. Additionally, 114 compounds were involved in the biosynthesis pathways of monoterpenoid, sesquiterpene and triterpenes, and $\alpha$-linolenic acids. Seventeen common compounds were involved in the metabolic pathways of isoprenes, fatty alcohols, and others. GC-MS metabolites from the five mangroves and their parts mainly differed in their isoprenoids or terpenoids; their number and content correlated with the tolerance level of mangrove species. In terms of volatile content and type differences, *A. ilicifolius* and *B. gymnorrhiza* were distinguished from the other three species, while leaf was clearly distinguished from the other parts, and the mangrove species had a greater factor effect than their part. Our study highlights the distribution and importance of secondary metabolites in mangroves and their role in mangrove tolerance. The study's findings will serve as a reference for future fundamental research on the molecular biology of mangrove plants and the development and use of pharmaceuticals.

## ACKNOWLEDGEMENTS

The authors would like to thank the editors and the anonymous reviewers for their constructive comments and suggestions, which helped to improve the quality of this paper.

### Funding

This work was supported by the Key-Area Research and Development Program of Guangdong Province (NO. 2020B020214001-ZKT03) and the Fundamental Research Funds for the Central Non-profit Research Institution of Chinese Academy of Forestry (CAFYBB2022MA005). The funders had no role in study design, data collection and analysis, decision to publish, or preparation of the manuscript.

### Grant Disclosures

The following grant information was disclosed by the authors:
Key-Area Research and Development Program of Guangdong Province: 2020B020214001-ZKT03.
Fundamental Research Funds for the Central Non-profit Research Institution of Chinese Academy of Forestry: CAFYBB2022MA005.

### Competing Interests

The authors declare there are no competing interests.

### Author Contributions

- Zhihua Wu conceived and designed the experiments, performed the experiments, authored or reviewed drafts of the article, and approved the final draft.
- Xiuhua Shang analyzed the data, prepared figures and/or tables, and approved the final draft.
- Guo Liu analyzed the data, prepared figures and/or tables, and approved the final draft.
- Yaojian Xie conceived and designed the experiments, authored or reviewed drafts of the article, and approved the final draft.

### Field Study Permissions

Field experiments were approved by Guangdong Provincial Science and Technology Department and Guangdong Provincial Forestry Bureau (project number: 20208020214001).

### Data Availability

  The compounds measured are listed in Tables S1, S2.
  The GC-MS data is available at the Dryad Digital Repository: Wu, Zhihua; Shang, Xiuhua; Liu, Guo; Xie, Yaojian (2022), Volatile compounds of five mangrove species and parts, Dryad, Dataset, https://doi.org/10.5061/dryad.ffbg79cz8.

## Supplemental Information

Supplemental information for this article can be found online at http://dx.doi.org/10.7717/peerj.15529#supplemental-information.

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
