# Peer review of "Comparative analysis of flavonoids, polyphenols and volatiles in roots, stems and leaves of five mangroves"

_PeerJ, doi:10.7717/peerj.15529_

## Round 0.1 · original submission · Major Revisions

Your manuscript was reviewed by three experts in the field. All the reviewers find the work interesting but raised multiple issues which need to be addressed properly. The reviewers provide detailed comments in their reviews and pointed out the areas where the manuscript needs to be improved. I also read the manuscript carefully and largely agree with the reviewers’ comments. The manuscript needs some major revision to meet the publication standard of PeerJ.

Reviewer 1 ·

Basic reporting

1. The manuscript was written in fluent English overall; however, there were some typographical and grammatical errors throughout the manuscript. Below are examples:
• In line 13, the s in “volatiles” is in bold
• In line 43, remove “[“. It’s grammatically unnecessary
• In line 152, change “shook” to “shaken”
To help the audience, I suggest scrutinizing the manuscript again to check these errors.

2. Introduction and background revealed context, and the authors explained the rationale for the study clearly, which is that identification of bioactive compounds in mangroves will be of interest for therapeutic purposes. The authors claim that the common metabolites found in the study were enriched in the terpene biosynthesis pathway. It would make the manuscript compelling if the authors commented on how these terpenes might have potential therapeutic benefits.

3. Some figures were too small and/or had poor resolution, and it was difficult to see them. Such examples include Figure 2, 5, 7, and 10. To help audience understand the data, I suggest the authors provide figures with better quality. Additionally, figure legends are difficult to see in Figure 12.


4. The overall structure of the manuscript conforms to PeerJ standards. One suggested change would be to move the paragraph from line 101 to line 112 to the methods section to describe the rational of using PLS-DA rather than in introduction.

Experimental design

5. The strength of the manuscript is that the study was designed well, and the GC-MS and statistical analyses were performed properly. These analyses can be very complicated sometimes, and I commend the authors for performing this properly.

7. The experimental design was clear with 5 different mangrove species being investigated in the study. One simple question that arose is, what was the rationale for the selection of these 5 different mangrove species? This was not clearly stated.

Validity of the findings

6. Thank you for providing the details for the mass spectrometry methods that were used in the study. One key issue here is how much confidence the authors have about the ID of the metabolites that were revealed in the study. The GC MS-QP2020 SE (Shimadzu) that was used for GC-MS analysis is a single quadruple mass spectrometer, and the authors compared the MS1 data to the NIST library to identify the metabolites. Metabolite identification is often a huge endeavor, and it requires either high-resolution mass spectrometry, fragmentation of analytes by MS/MS analysis, or running standards to compare chromatographic profiles, none of which was performed in the study. Since the conclusion of the study, that the common metabolites identified in the 5 mangroves are enriched in the terpene biosynthesis pathway, is based on the ID of the metabolites, the weight of evidence of the findings is unclear. It would help to include some example chromatograms from the NIST library and compare them to the original data to see how they align.

Additional comments

Taken together, the manuscript would satisfy the editorial criteria for basic reporting and experimental design after making the suggested changes mentioned above. However, the most important issue for this manuscript is validity of the findings. As stated in item 6, using the mass spectrometry methods that the authors used for the study, it would be extremely difficult to ID the metabolites with high confidence, and insufficient data were provided to support their identification efforts. Without clarifying this critical point, the conclusions of the manuscript would not be substantiated.

Annotated reviews are not available for download in order to protect the identity of reviewers who chose to remain anonymous.

Reviewer 2 ·

Basic reporting

PEERJ
Reviewer’s Report
TO AUTHORS

MS#: #80417
TITLE: Comparative metabolomic analysis of volatiles in roots, stems and leaves of mangroves reveals differences in terpene synthesis.
AUTHOR(S): Zhihua Wu , Xiuhua Shang , Guo Liu , Yaojian Xie

Comments to authors:
In the current manuscript, the authors set out to identify volatiles in five mangrove plant species. Samples were collected from leaf, root, and stem. They then extracted total metabolite from mangrove plants, and analyzed volatile content by GC-MS followed by data analysis. The authors then compare the identified metabolite with different species of Mangrove and different plant part sample. Manuscript is as such well-designed experimentally and analyzed but manuscript is too descriptive in nature.
Abstract Section: Authors should take the catalogue of metabolites number to result section, it should be deleted from the abstract section.
Line 18-21: The number of compounds found was from 41 to 86 in each part of five mangrove species, A. corniculatum leaves and A. marina roots contained a maximum of 86 compounds. 247 compounds were found in A. corniculatum, 245 in K. candel, and 240 in A. marina. Roots, stems, and leaves each had 399, 342, and 339 compounds. There were 40 unique compounds in A. corniculatum leaves, and 39 in A. corniculatum stems.
Introduction section: It is too descriptive about metabolomics and GC MS; my suggestion is to rewrite these sections and concise it.
Line 78-89. For GC-MS description
Line 90 -100: for Metabolomics Description
Line 101-112: PLS-DA description should be in appropriate section; my suggestion is to take it data analysis or the result section, where authors describe about PLS-DA analysis.
Line 113: Since 2010, several review papers have summarized bioactive compounds derived from true mangroves and semi-mangroves worldwide (Wu et al., 2008; Li et al., 2009). Author’s should correct the statement according to the reference or vice versa
Discussion:
Line 325- 337 discussion about GC-MS too detailed and it’s about GC-MS technique, authors should restrict discussion part accordingly to their result.
Figure: Figure 5 is too small; Figure 10 color combination is not good as all species is not clear and Figure 11 quality is also not good. Author’s advised to export the figure from software at high resolution and or in pdf format.
Minor issues:
1. Line 13 Abstract section: Voltiles “S” is in bold form
2. Line 31 “etc”. should be omitted
3. Authors should restrain themselves to make general statement
e.g., Line 51. The interest of scientists in plant extracts and compounds isolated from natural materials has recently become a trend.
4. Line 58: Incorporate reference for antibacterial effects.
5. Line 167: Scan “S” should be in lower case
6. Author’s should write the Genus name in full form at first place it is used in the Manuscript.
7. Line 139 Delete “content” word
8. Line 155 Delete “condition”word
9. Line 316: ORA should be deleted
10. Supplement table AI_leaf_21 is not in proper format,
11. Supplement Data_for_figure5_to_figure8_71_commond_compounds; “Commond” should be Common.

Experimental design

Manuscript is as such well-designed experimentally and analyzed.

Validity of the findings

Discussion part needs modification

Reviewer 3 ·

Basic reporting

This study describes GC-MS chemical profiling of the different parts (roots, stems, leaves) of 5 mangrove species and subsequent analysis of the data on a metabolomics platform. While the authors are commended for the extensive data collected, the experimental design as well as data presentation and analysis require substantial improvement. The focus of this work is unclear. There is a section on polyphenols and flavonoids which is unrelated to the main focus on volatiles (=The title does not match the content of the manuscript). The discussion is weak with very little to no attempt to analyse the data from this study in the context of what is known in the literature.

1. The Introduction section is unnecessarily long. Instead of proving factual account on natural products, GC-MS, metabolites and PLS-DA separately, the authors are encouraged to emphasise on the background of this research and to highlight the research gap.
2. The objectives need to be clear.
3. Provide justification for the selection of 5 mangrove species used in this study.
4. Lines 205-209: Unnecessary information could be omitted.
5. Line 233: what is chemical composition compound?
6. The section on "evaluation of GCMS method" is unnecessary as optimization and method development was not undertaken in this study. That section contains mostly factual account on GCMS which in my opinion can be removed.
7. The lengthy elaboration on phenolics and flavonoids does not include discussion on the results obtained in this study, and this section is irrelevant to the research objectives too.
8. Similarly, the discussion on the mangrove volatiles does not take into account the data from this study. The roles of the volatiles, as stated, is not supported any data.
9. Please revise the title to better reflect the content of this manuscript. There is very little coverage on the involvement of metabolites from the biosynthesis perspectives.
10. The quality of some images are very poor.

Experimental design

The overall experimental design needs to be clarified.

Main comments:
The section of estimation of polyphenols and flavonoids are completely irrelevant to the analysis of volatile compounds.

Standard methodologies were used but there is insufficient information in some sections.
Line 127: Please state the name of the person who identified the seedlings.
Lines 139-147: Please provide a reference for the methods used to determine the polyphenol and flavonoid content. How did the authors prepare the samples for both assays?
Line 142: Please indicate the source and purity of catechin.
Line 146: Please indicate the source and purity of gallic acid.
Line 151: The choice of extracting solvent is not suitable. If volatiles are the target, why did the authors use 70% ethanol instead of solvents like hexane, acetone, etc.?
Line 170: The same sample was analysed (injected into GCMS) three times?
Line 174: Please justify 90% similarity as the cut off point.

Validity of the findings

1. Identification of compounds via library matching is not convincing, especially this is not supported by mass fragmentation data. Instead, appropriate standards should be used for confirmation.

2 There is insufficient analysis to look into the relationship in chemical profiles with different parts of mangrove seedlings and in different species which is supposedly the study's main focus.
- What can be deduced regarding the differences in the distribution of compounds in different parts of the plants?
- Do the different species show similar profiles/trends? What are the possible reasons?

3. Data from the metabolomics platform are also inadequately discussed. Need to provide more description and interpretation of the output from metabolomics platform.

4. The conclusion should be precise and linked to the research questions/objectives. What can be concluded from the findings using 5 different species?

Additional comments

The authors presented data from spectrophotometric and GC-MS analyses to unravel the chemical composition of the different parts of 5 mangrove species. The same data set, however, do not provide insights into the variation in different parts of the mangrove and the different species, let alone to "reveal differences in terpene synthesis" that was only briefly discussed in the entire manuscript. Substantial improvements, including reorganisation of the data, are recommended.

---

## Round 0.2 · Major Revisions

I agree with reviewer 1 who pointed out valid concerns therefore your manuscript needs further revision.

Reviewer 1 ·

Basic reporting

As stated in the previous review, the manuscript was written in fluent English. I thank the authors for providing answers to the questions raised in the previous review.

Experimental design

I thank the authors for providing more details on the GC-MS analysis and calibration curve of standard solutions for GC-MS. The generation of calibration curves for linolenic acid, octadecanoic acid, and linoleic acid shows that the GC-MS instrument used can accurately produce data in the range of dilutions between 0.1 and 0.5 mg/mL. This dynamic range is extremely narrow for a mass spectrometry instrument. My main concern in the last review was that the methods used in the study are insufficient to identify the metabolites with confidence. The thesis of the manuscript and data analyses are based on the assumption that the authors have confidence in the identification of the metabolites. Generation of calibration curves for these fatty acids does not increase the confidence in identifying the metabolites directly. The authors provided a reference from Wei et al, 2014 to show that mass spectrum matching is a widely used approach for compound identification in GC-MS. After reviewing the paper from Wei et al., 2014, the paper provides a theoretical foundation for this method; however, whether or not this method is practical is highly questionable. As reviewer 3 also pointed out, authentic identification of metabolites requires fragmentation studies to compare fragmentation patterns to standards as well as comparing the retention times of authentic standards, neither of which were carried out in the study. These are the gold standard for metabolite identification, and even performing a such study on a few metabolites, say from C10 and C15 isoprenoids, would strengthen the arguments of the study. This was, however, not carried out, and I would love to be supportive of the authors for their study, but without these steps, the findings of the study are speculative, and the weight of evidence is not sufficient to arrive at the conclusions that the authors drew.

Validity of the findings

As mentioned in the experimental design section, because of the lack of rigor in performing experiments to identify the metabolites, the validity of the findings is still unclear. To strengthen the arguments of the manuscript, I suggest the authors to purchase some standards for the key metabolites found in the study (such as those in the terpene synthetic pathway), run them in GC-MS and compare the retention time, and if possible, perform additional GC-MS/MS studies to compare the fragmentation patterns of the analytes. These steps are crucial to draw the scientific conclusions that the author provided.

Reviewer 2 ·

Basic reporting

No comment

Experimental design

no comment

Validity of the findings

no comment

Additional comments

Authors considered the Reviewer's suggestions positively and manuscript is revised accordingly.

---

## Round 0.3 · Minor Revisions

Please address the concerns raised by the reviewer.

Reviewer 1 ·

Basic reporting

No issue except in the revised version of Fig. 5 title, it says 'retent time' instead of 'retention time'. This is a minor typo, but please fix this before publishing.

Additionally, the new mass spectrometry figures contain many labels in the Chinese language, which cannot be understood for non-Chinese speaking audience. To help the future audience better, please convert the Chinese labels into English. I understand that for each page, there is a legend that explains which Chinese word corresponds to English words, but it is extremely difficult to understand.

Experimental design

I thank the authors for providing additional experimental data to increase the confidence in the GC-MS data. The purchase of the authentic standards and comparison of their retention time to the original data strengthen the authors' original claim.

Validity of the findings

It is convincing that the retention times of the purchased standards were close to the previous data, and most of them were shifted by a similar value around 0.5.

Overall, the supplemental data provide additional confidence in interpreting the authors' original data. I thank the authors for their persistent efforts in providing these data.

---

## Round 0.4 · accepted · Accept

I appreciate the authors' efforts in revising the manuscript substantially. The current version is suitable for publication in PeerJ.